# Extreme trait GWAS (Et-GWAS): Unraveling rare variants in the 3,000 rice genome

Niranjani Gnanapragasam[1], Vinukonda Vishnu Prasanth[1], Krishna Tesman Sundaram[1], Ajay Kumar[1], Bandana Pahi[1], Anoop Gurjar[2], Challa Venkateshwarlu[1], Sanjay Kalia[3], Arvind Kumar[2], Shalabh Dixit[4], Ajay Kohli[4], Uma Maheshwer Singh[2], Vikas Kumar Singh[1], Pallavi Sinha[1]

Identifying high-impact, rare genetic variants associated with specific traits is crucial for crop improvement. The 3,010 rice genome (3K RG) dataset offers a valuable resource for discovering genomic regions with potential applications in crop breeding. We used Extreme Trait GWAS (Et-GWAS), employing bulk pooling and allele frequency measurement to efficiently extract rare variants from the 3K RG. This innovative approach facilitates the detection of associations between genetic variants and target traits, concentrating and quantifying rare alleles. In our study, on grain yield under drought stress, Et-GWAS successfully identified five key genes (*OsPP2C11*, *OsK5.2*, *OsIRO2*, *OsPEX1*, and *OsPWA1*) known for enhancing yield under drought. In addition, we examined the overlap of our results with previously reported *qDTY*-QTLs and observed that *OsUCH1* and *OsUCH2* genes were located within *qDTY2.2*. We compared Et-GWAS with conventional GWAS, finding it effectively capturing most candidate genes associated with the target trait. Validation with resistant starch showed similar results. To enhance user-friendliness, we developed a GUI for Et-GWAS; https://et-gwas.shinyapps.io/Et-GWAS/.

## Introduction

With the current advances in DNA sequence technology, in the past two decades, there was a rapid acceleration in the genomic resources of food crops and in the development of tools and technologies that use this vast amount of genomic sequence information for the crop improvement (Bohra et al, 2020; Thudi et al, 2021). Re-sequencing of 3,010 rice genomes (3K RG) provided an avenue to use all the natural variation that exists in cultivated and wild rice and use that information to identify genes and genomic regions that can be used to drive the next generation tailor-made crops (Wang et al, 2018). The genome information will also be very much useful to identify the association between structural variants and trait phenotypes (Żmieńko et al, 2014). Genome-wide association study (GWAS) is known to be a powerful tool to detect the genetic variations associated with the targeted traits in crops (Huang & Han, 2014). These variants are then used to track and identify causative genes. Hence, addressing the limitations within the conventional GWAS has also become a hot topic in the post-genomic era (Korte & Ashley, 2013; Tam et al, 2019).

One of the significant limitations of GWAS is that the variants identified through it can only explain a certain level of heritability of the trait (Zuk et al, 2012). Furthermore, GWAS uses stringent thresholds in data preprocessing (filtering) and association analysis (multiple-test correction, Bonferroni correction) to reduce false positives. Most crop traits are complex, where multiple quantitative trait loci of genes present throughout the genome affect the phenotypic expression in various quantities. While studying such traits through GWAS, high-impact rare variants are thrown into false negatives because of the higher genome-wide cutoff value (Dickson et al, 2010). Though missing heritability has various factors influencing GWAS, such as gene-to-gene and gene-to-environment interactions, a certain level of improvisation by incorporating the rare SNPs can be done (Futschik & Schlötterer, 2010). New statistical method could be developed that reduces the number of tests conducted and so increases the *P*-value of the Bonferroni correction (Wang et al, 2016; Hamazaki & Iwata, 2020). This can also be done by developing a sampling strategy of the population/variants where the local sampling or the contrasting phenotypes are considered (Lorenz et al, 2010; Gyawali et al, 2019; Alseekh et al, 2021).

It was reported that if we combine the bulk segregant analysis (BSA) sampling strategy with the GWAS, where selective genotyping of the individuals with contrasting phenotypic performance can effectively map the genetic variants associated with the trait of interest (Michelmore & Kesseli, 1991). BSA has been successfully used in QTL mapping of biparental or multi-parental segregant

[1]International Rice Research Institute, South Asia Hub, Patancheru, India   [2]International Rice Research Institute, South-Asia Regional Centre, Varanasi, India   [3]Department of Biotechnology, CGO Complex, New Delhi, India   [4]International Rice Research Institute, Los Banos, Philippines

Correspondence: p.sinha@irri.org; v.k.singh@irri.org
Arvind Kumar's present address is International Crops Research Institute for the Semi-Arid Tropics, Patancheru, India

populations, which has equal representation of the parental genotype except in the regions that are associated with the trait of interest (Takagi et al, 2013; Singh et al, 2017). Even the natural population containing individuals of diverse genetic backgrounds can be used as a segregating mapping population where the historical recombination events could help associate the variants to the trait of interest (Yang et al, 2015; Li et al, 2019). Opposed to the biparental segregating population, the diversity panel from the natural population will have diverse variation for all the traits. However, when it is phenotyped under controlled conditions, it will ensure that the contrasting phenotypes will possess significant genomic variation for the trait of interest while having random variation for nontarget traits.

Pooled-based GWAS using extreme phenotypes have been applied in both plant and human genetics, with various techniques such as Pool GWAS, XP-GWAS, and pooled sample–based GWAS. However, these studies typically involve the pooling of DNA samples from multiple plants (or individuals) and sequencing them together, as seen in previous works (Gaj et al, 2012; Yang et al, 2015; Lirakis et al, 2022). When researchers aim to leverage existing sequence data, such as the 3,000 Rice Genome (3K RG) dataset, it necessitates bioinformatics skills, particularly proficiency in coding. This requirement for coding expertise can pose a significant challenge for breeders and researchers. As a result, this technical hurdle can be a notable barrier to conducting these studies, particularly within the field of plant breeding.

Based on the abovementioned context, we have successfully developed the extreme trait GWAS pipeline (Et-GWAS) to unravel rare genetic variants using the comprehensive publicly available 3K RG variant data. The Et-GWAS tool is equipped with the entire 3,010 genome variant dataset, conveniently packaged with a graphical user interface (GUI). This architecture is designed to streamline the trait mapping process, requiring only the phenotyping data of the target trait(s) from the 3K RG panel. In our research, we employed Et-GWAS to effectively map grain yield under reproductive stage drought stress, leveraging the power of rare variant discovery within the 3K RG dataset. In addition, we thoroughly validated the functionality of the Et-GWAS tool by applying it to an extended study of the resistant starch (RS) trait in rice.

To ensure the reliability and accuracy of our Et-GWAS pipeline, we compared its results with those obtained through conventional GWAS. Through this, we confirmed the robustness and efficacy of our developed architecture, demonstrating its ability to accurately identify and characterize rare variants associated with the target traits. The Et-GWAS pipeline represents a significant advancement in trait discovery, offering a user-friendly and efficient approach to leveraging the wealth of genetic information in the 3K RG dataset. Its versatility extends beyond rice, as it can be seamlessly adapted for use in other crops and species, facilitating trait-mapping studies and accelerating progress in crop improvement and agricultural research.

The haplotypes that perform well in different conditions were identified and the corresponding donor lines for the drought resistance are also reported here. In this process, we also developed a GUI that will use the genomic information available in open source and identifies the top-performing haplotypes of the drought-responsive genes.

## Results

### Principle of Et-GWAS pipeline

The Et-GWAS method leverages historical recombination events within a diversity panel to identify rare, high-impact variants associated with the targeted traits. Inspired by the extreme phenotype GWAS approach, it combines BSA with GWAS, treating the diversity panel as a segregating population (Yang et al, 2015). The method uses available genome sequence information to construct extreme pools and a random bulk, representing the genomic variation across the entire diversity panel. The contrasting germplasms are grouped, and allele frequencies are measured to facilitate the detection of marker–trait associations (MTAs). To illustrate the Et-GWAS, we presented a schematic representation in Fig 1A–C, focusing on its application to yield under reproductive stage drought in rice. The Et-GWAS pipeline, along with the associated ShinyApp, is accessible at https://github.com/IRRI-South-Asia-Hub/Et-GWAS, offering a user-friendly and efficient tool for researchers to discover and analyze rare variants associated with various agronomic traits. The analysis involves the following three key steps:

(1) Sampling: Constructing the diversity panel and bulk preparation based on the trait distribution.
(2) Pooling: Combining the sequence data from the bulks and quantitative measurement of pooled allele frequency.
(3) Screening: Association analysis and identification of donors with top-performing haplotypes.

The diversity panel for Et-GWAS should be a representative subset of crop germplasm, using publicly available sequence information to encompass most random variants in the population. After careful phenotyping of the panel under specific environmental conditions for the target trait, the distribution of phenotypic values helps in selecting the bulks for analysis. Bulk size determination relies on the homozygosity of the lines and sequence depth. For natural populations with segregation, a larger bulk size is required (should contain 70–90% heterogeneity of the population). The Et-GWAS process involves generating three bulks from the diversity panel. Contrasting bulks, namely the low and high bulks, are selected from the lower and upper sides of the population, respectively. An equal number of lines are randomly chosen from the diversity panel to form the control bulk. The control bulk serves as a comparison group for the extreme bulks, assisting in eliminating alleles that show significant differences between the extreme bulks but are not associated with the target trait. This stringent approach enhances the precision and reliability of identifying trait-associated variants within the diversity panel.

The variant information of the lines from each bulk is pooled to quantify the reference and alternate alleles present at each polymorphic position. Hence the input file for the association study will have the SNP information and the allele frequencies of reference and alternate alleles from the high, low, and random bulks in an order (refer Table S1 for the file format). These SNPs are then filtered for MAF (0.05) but not for the

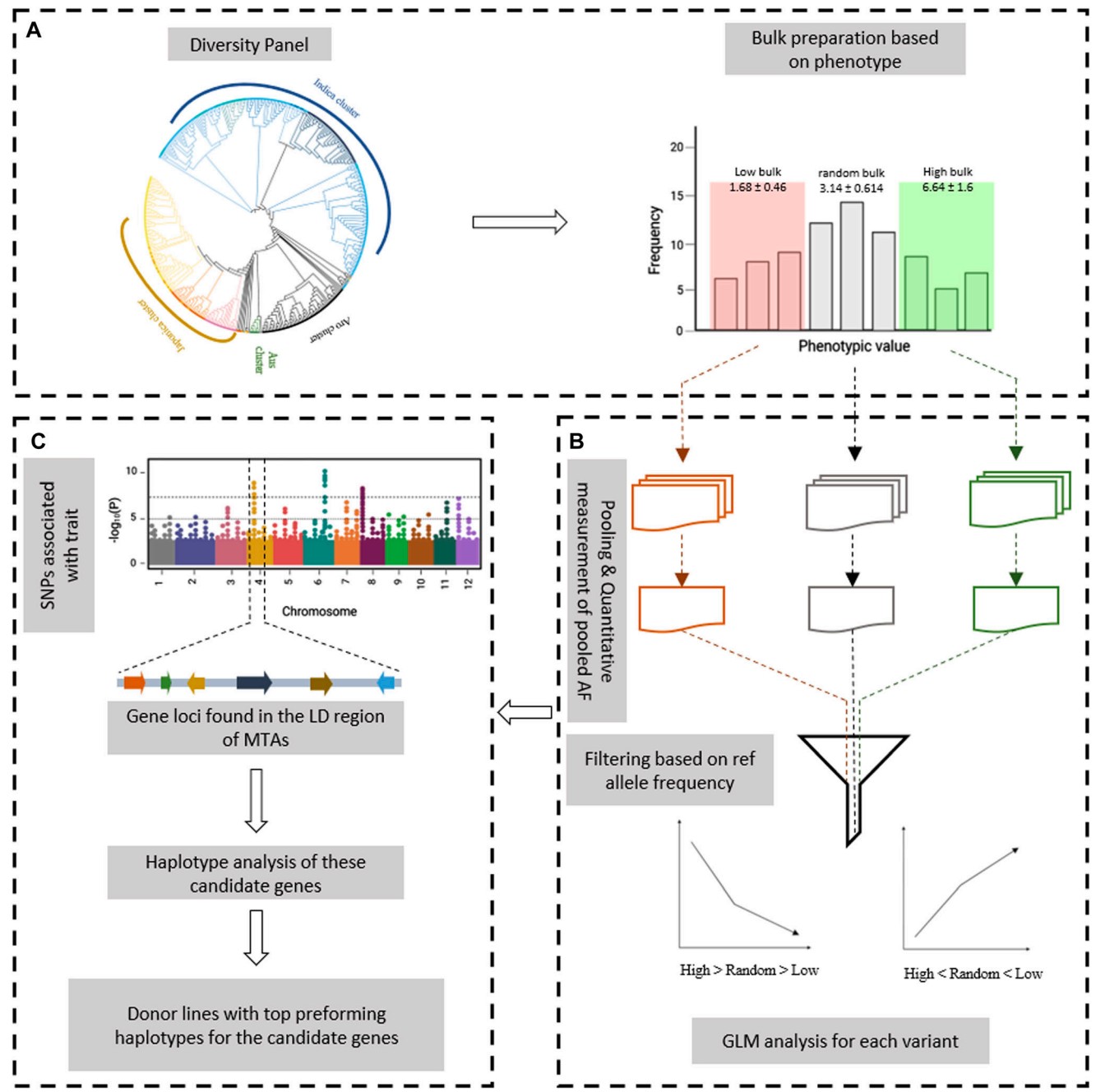

**Figure 1.   The schematic representation of the Et-GWAS pipeline.**
**(A, B, C)** The workflow consists of three steps; (A) sampling, (B) pooling of variant information from available sequence data, and (C) association analysis and identification of donors with top-performing haplotypes.

missingness, because the method targets the rare variants that are present in low count in each bulk. In addition, density or coverage is employed as a secondary filter, aiding in reducing experimental errors by pooling the sequence information of the bulks.

The selected SNPs are analysed using the general linear method (GLM), considering the low, random, and high bulks as linear sequences. Significant monotonous trends exhibited by certain SNPs are considered markers associated with the trait of interest. Subsequently, candidate genes are scanned based on the crop's linkage disequilibrium (LD). The pipeline further identifies the candidate genes' top-performing haplotypes and the lines that harbour one or more of these top-performing haplotypes (see the Materials and Methods section).

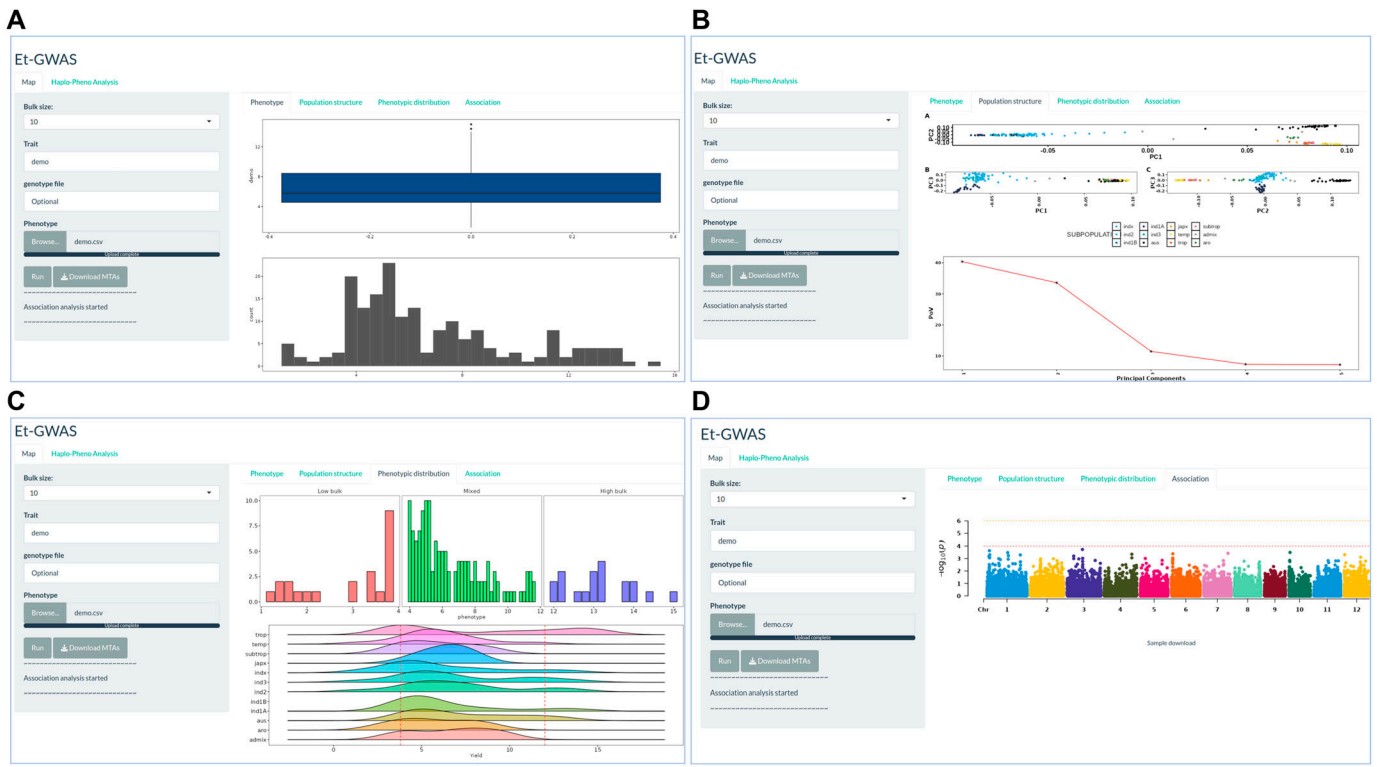

**Figure 2. Main interface of the Et-GWAS application.**
Screenshots of panels for the main tabs are shown. **(A)** The "Phenotype" tab displays the phenotypic distribution of the given panel. **(B)** The "Population structure" tab allows users to visualize the subpopulation structure of the panel. By default, three principal components are calculated. **(C)** The "Phenotypic distribution" tab allows users to visualize the phenotypic distribution in each bulk. The users have the flexibility to choose the bulk size according to their panel size. **(D)** The "Association" tab generates Manhattan plot with two significant values ($10^{-4}$ & $10^{-6}$). Rest of the plots and tables will be generated and kept in a zip file which users can download using the right panel "Download" button at the end of the analysis.

## Graphical user interphase and stand-alone package of Et-GWAS

The Et-GWAS application is user-friendly and requires no prior knowledge of R programming. It operates through a GUI that guides users through the process. To begin, users input the desired bulk size, trait name, and phenotypic file on the starting page of the application. Comprehensive instructions regarding data formatting can be found in the documentation, ensuring smooth data input. The functionality of the pipeline and test results are displayed in Fig 2A–D; demonstrating its effectiveness in trait association studies. It is important to note that the Et-GWAS application hosted on the server has a limit of handling 30K markers.

To accommodate higher marker coverage, we recommend running the application locally from GitHub. This allows for a broader marker range, enhancing the precision and scope of trait discovery and association analysis. For a smooth run, it requires a workstation with the memory of 64 GB or above and a CPU speed of 3.0 GHz or above.

## Proof of concept

Et-GWAS was assessed for grain yield under reproductive stage drought-stress conditions in rice. Drought exerts a significant negative impact on yield at all developmental stages, with reproductive stage stress causing the most severe damage. Our analysis centers on two-season phenotypic data from the 3K subset, which underwent reproductive-stage drought stress conditions.

## Establishing the diversity panel and population structure

The drought stress diversity panel (3K Subset) comprises 399 early and medium maturity accessions from the 3K RG panel. These accessions offer great potential for breeding programs, and their high-quality sequence data are available (https://snp-seek.irri.org/). The LD-pruned 3K RG 1M GWAS SNP Dataset, encompassing all chromosomes, was downloaded for analysis. The filtered dataset resulted in a final set containing 600,288 nonredundant SNPs (see the Materials and Methods section for details). The density and distribution of alleles within various fractions of the genome of the working dataset are summarized in Fig S1A and B. The average marker density per Mb of the genome was 503 SNPs, with the highest density observed on Chr 3, at 416 SNPs/MB. Intergenic regions accounted for only 33% of the SNPs, whereas exonic SNPs represented a mere 3%. Over 50% of the SNPs were found in the upstream and downstream regions of the genic regions.

The principal component analysis (PCA) of the working dataset distinctly reveals three clusters: cluster I comprises indica subpopulations (*ind1*, *ind1A*, *ind1B*, *indx*), cluster II contains *aus* subpopulation, and cluster III comprises japonica subpopulations

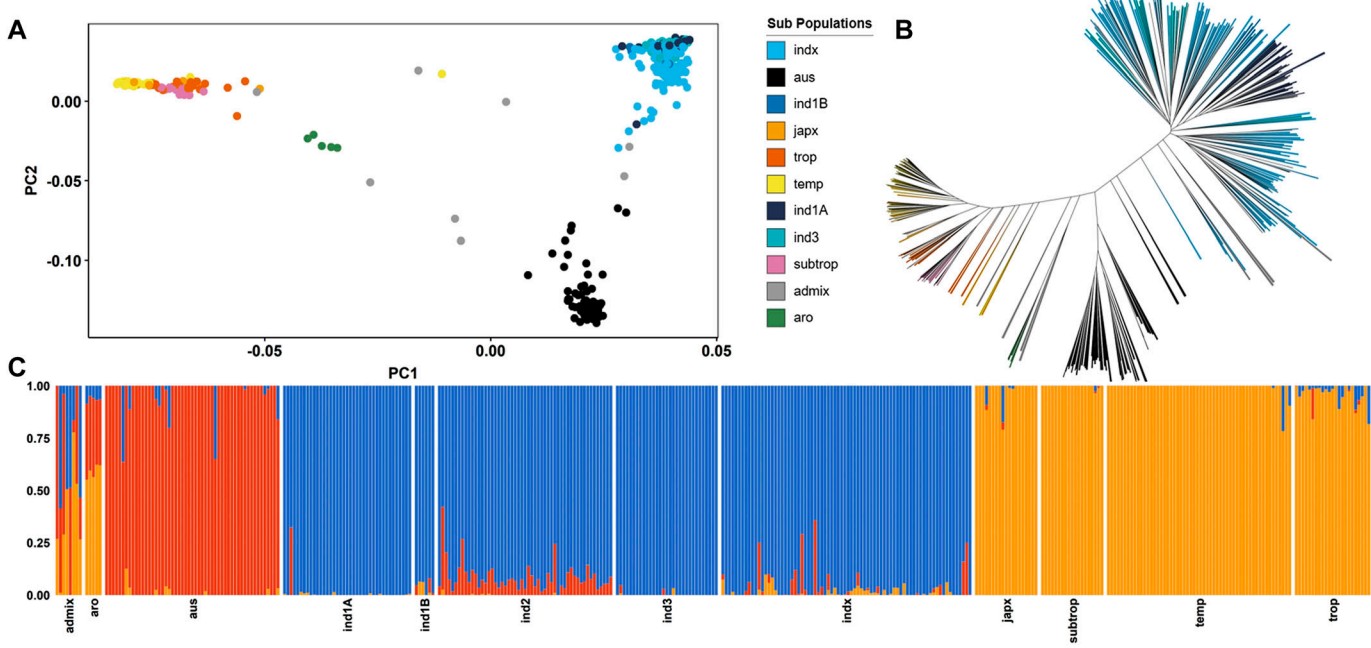

**Figure 3. Population structure.**
**(A)** The principal component analysis shows the clear segregation of various subpopulations. **(B)** The same result was observed on the dendrogram of the diversity panel. **(C)** The ancestry proportions through clusters (K = 3) showing three clusters corresponding to *indica, aus,* and *japonica* as established by 3K genotypic data.

(*japx, trop, temp, subtrop*). The first PC accounts for half of the variance and effectively distinguishes all three clusters, with clusters I and II exhibiting a close relationship. The second PC differentiates cluster II from cluster III, explaining 30% of the variation (Fig 3A). In the PCA, the *aro* subpopulation remains in proximity to the *japonica* cluster, whereas the third PC separates it from the other clusters. As expected, the *admix* subpopulation occupies an intermediate position between the clusters. The phylogenetic diversity, represented by the neighbor-joining tree, aligns with the PCA results, as the accessions cluster according to their corresponding subpopulations (Fig 3B). As anticipated, the optimal K value of three effectively separates the accessions (Fig 3C) and maximizes the likelihood of the genomic dataset.

## Phenotypic variations in the population

For method evaluation, we used single-plant yield (SPY) data (average of five-plant) under reproductive stage drought stress conditions from the dry seasons of 2019 (DS2019) and 2020 (DS2020). The drought stress conditions in both seasons significantly differed because of the impact of the pandemic and unforeseen rains (Fig S2A and B). The density of the bulks and subpopulation-wise contribution indicate distinct populations for the two-season data, leading us to analyze them separately (Fig S3A–C). The distribution of SPY data from these two seasons showed significant difference in the Kolmogorov–Smirnov test, as depicted in Fig S3C. The SPY under drought showed variability, ranging from 0.6 to 18.3 g in DS2019 and 0.4 to 16.5 g in DS2020. The average SPYs of the *Indica* and *Japonica* subpopulations displayed consistent performance in both seasons, with a substantial

proportion of drought-tolerant lines (Fig S3A). Regarding drought susceptibility, the drought susceptibility index (DSI) in all subpopulations spanned from −0.32 to 1.2 in DS2019 and −0.07 to 1.2 in DS2020, with a right-skewed distribution (Fig S3B). This wide range of DSI values across different subpopulations highlights significant genotypic variability associated with grain yield under drought stress conditions.

## Identification of MTAs through Et-GWAS

We conducted an association analysis of pooled allele frequency with grain yield under drought stress conditions, considering various parameters: (a) pool size, (b) filtering levels, and (c) inclusion of subpopulations with skewed phenotypic distribution. A pool size of 20% with minimum filtering, which considered only MAF and read depth, while including all subpopulations yielded comparably good results, identifying a high number of SNPs within the reported QTLs associated with grain yield under drought stress conditions. Because the diversity panel represents a highly segregated population, smaller bulks could not adequately cover many essential SNPs. For the 20% pool size, we provide the sample size and average missingness in Table 1, and the contribution from different subpopulations is illustrated in Fig S4. Et-GWAS successfully identified 28 MTAs associated with grain yield under drought conditions in the DS2019 data, distributed across chromosomes 1, 2, 3, 5, and 6 (Table S2). Similarly, using the DS2020 data, 26 MTAs were identified, spanning the genome except for chromosomes 3 and 7 (Table S3). The phenotypic variance explained (PVE) by the MTAs ranged from 3.91% to 4.77% for DS2019 and 3.57% to 4.43% for DS2020 (Fig 4A and B).

**Table 1.** Summary statistics for phenotypic and genotypic data of the bulks used for Et-GWAS under two seasons.

| | Number of lines in bulk | | Mean SPY (g)± sd | | Coverage | Avg. Alt allele frequency | |
|---|---|---|---|---|---|---|---|
| | DS2019 | DS2020 | DS2019 | DS2020 | | DS2019 | DS2020 |
| Low bulk | 76 | 78 | 1.68 ± 0.46 | 1.01 ± 0.23 | | 0.22 (0.006-0.67) | 0.22 (0.006 − 0.7) |
| High bulk | 76 | 78 | 6.64 ± 1.6 | 2.72 ± 0.86 | ~20% | 0.23 (0.006-0.66) | 0.22 (0.006 − 0.65) |
| Random bulk | 76 | 78 | 3.14 ± 0.61 | 7.32 ± 2.08 | | 0.217 (0.008-0.69) | 0.21 (0.0076 − 0.69) |
| Diversity panel | 378 | 389 | 3.5 ± 1.89 | 3.3 ± 2.4 | | | |

Bulk size—20% of the diversity panel. Mean SPY ± Sd—mean and standard deviation of single plant yield (in grams). Avg. Alt allele frequency—average alternate allele frequency (it is average minor allele count by considering all the SNPs and all the lines present in each bulk).

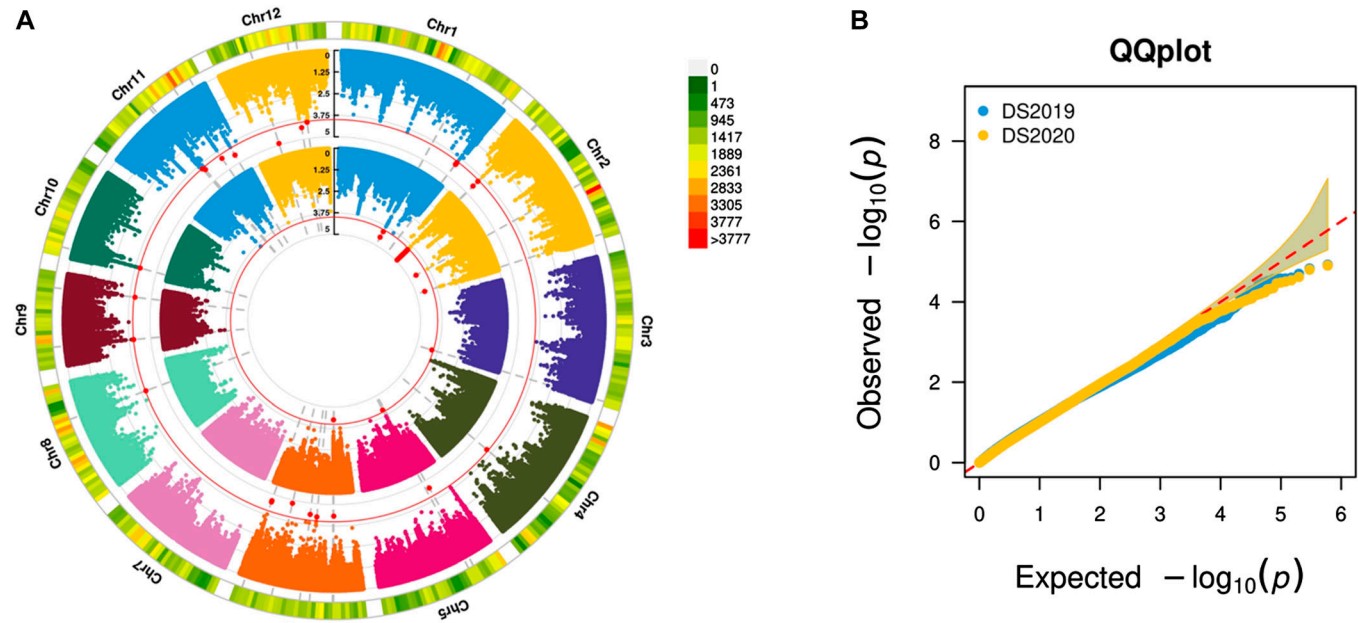

**Figure 4. Circular Manhattan plot and Q–Q plot for single plant yield under drought.**
**(A)** Et-GWAS for season DS2019 (inner circle) for season DS2020 (outer circle) are shown. The genome-wide threshold of $10^{-4}$ (red line) was used for the identification of significant SNPs. The outermost ring depicts the SNP distribution of the 600288 SNP working set. **(B)** The Q–Q plot shows the performance to predict true positives from both the seasons' data.

We precisely located the identified MTAs within the rice genome using the RAP database. Among the 54 MTAs, 17 were found within the genic regions (Table 2), whereas the remaining MTAs were in the intergenic regions. Notably, two significant genes were associated with grain yield: *OsNPC2* (non-specific phospholipase C), involved in phospholipid signalling and influencing plant growth under stress conditions, and *OsmiR156d*, a microRNA acting as a genetic regulator for various biological processes. In addition, other loci with MTAs were linked to yield-related traits and DNA repair processes. Furthermore, we examined the consistency of MTAs between the two seasons and identified 16 overlapping MTAs n the Chr 2 region between the DS2019 and DS2020. Interestingly, the *OsGL1-2* gene harboured two SNPs associated with grain yield under drought stress conditions, with each SNP identified from different seasons. The PVE of these SNPs was ~3.9% and 4%, indicating their significant impact on the trait.

## Unraveling the candidate genes using Et-GWAS for yield under drought stress in rice

We identified a total of 141 candidate genes from the MTAs identified in both the DS2019 and DS2020 datasets. The list of locus names along with their functions is presented in Table S4. Notably, 16 of these loci have been previously reported for their involvement in drought stress tolerance through various mechanisms. Among the candidate genes, there are genes that are known for their role in drought response such as notable transcription factors (SRS family, bHLH TF, WRKY), genes involved in biosynthesis processes (glycosylt, cuticular) and stress-responsive proteins (U-box, MAP kinase). The list of the drought responsive genes identified are given in Table S5. Further functional characterization revealed the details of candidate genes known to be involved in drought tolerance. The gene ontology analysis of these genes demonstrated their

**Table 2. List of genic MTAs identified in the Et-GWAS approach peak SNP: position of MTA in basepair.**

| Loci name | Gene function | Chr | Gene interval (bp) | Peak SNP | SNP | PVE (%) |
|-----------|---------------|-----|--------------------|----------|-----|---------|
| *Os01g0955000* | Phosphoesterase family protein (*OsNPC2*) | 1 | 42063261–42067784 | 42064340 | G/A | 3.11 |
| *Os02g0179250* | Hypothetical protein | 2 | 4394882–4398791 | 4395405 | C/T | 3.35 |
| *Os02g0179300* | Similar to RAD23 protein | 2 | 4407991–4412772 | 4408874 | C/A | 3.03 |
| *Os02g0180100* | Ubiquitin carboxyl-terminal hydrolase 1 family protein | 2 | 4455016-4457715 | 4455865 | A/G | 5.75 |
| *Os02g0180200* | Control of tiller angle and Regulation of shoot gravitropism | 2 | 4460342–4463571 | 4461960 | T/G | 5.57 |
| *Os02g0180800* | Primary microRNA of *OsmiR156*. Affects tillering. | 2 | | 4501241 | C/T | 5.67 |
| | | 2 | | 4501272 | G/A | 5.19 |
| | | 2 | 4500090–4513333 | 4508488 | T/A | 5.39 |
| | | 2 | | 4513103 | A/T | 5.91 |
| *Os02g0812600* | F-box domain containing protein. Involved in protein–protein interaction. | 2 | 34798281–34801473 | 34799960 | A/G | 5.63 |
| *Os04g0451100* | Similar to H0815C01.3 protein | 4 | 22472184–22479333 | 22474294 | T/A | 3.04 |
| *Os06g0244100* | Tetratricopeptide-like helical domain containing protein | 6 | 7488066–7496627 | 7488181 | G/A | 3.23 |
| *Os09g0266300* | Conserved hypothetical protein | 9 | 5005986–5006587 | 5006508 | A/G | 2.98 |
| *Os11g0657400* | Involved in lignin biosynthesis and deposition. (*OsPEX1*) | 11 | 26317273–26321503 | 26320189 | G/A | 3.36 |
| *Os12g0264500* | Similar to CoA-thioester hydrolase CHY1 | 12 | 9350914–9365572 | 9359783 | G/T | 3.70 |

Chr, Chromosome; SNP, SNP change observed in the genotypes; PVE, phenotypic variation explained; genome-wide *P*-value cutoff used: $10^{-4}$

involvement in molecular activities associated with drought stress tolerance (Fig S5). This comprehensive identification and characterization of candidate genes shed light on the genetic basis of drought tolerance in rice.

## Superior haplotypes of candidate genes associated with yield under drought stress

Genes or loci harbouring MTAs and displaying linkage disequilibrium (LD) with the identified MTAs are selected for further *Haplo-Pheno* analysis, where the phenotypic performance of haplotypes from the candidate genes is evaluated. The diversity panel contains varying numbers of haplotypes, ranging from 3 to 35, and multiple loci exhibit haplotypes with a wide range of phenotypic values. These results highlight the impact of epistatic interactions, which can either enhance or mask the effects (positive or negative) of the haplotypes. The *Haplo-Pheno* analysis identified superior-performing haplotypes from four genes that consistently performed well across both seasons. In addition, six genes have superior-performing haplotypes only in DS2019, whereas two genes possess superior-performing haplotypes solely in DS2020.

The detailed information of the haplotypes for consistent and season-specific haplotypes along with their average phenotypic performance are documented in Tables S6 and S7, respectively. Among these haplotypes, several transcription factors (iron-related bHLH) known for their role in the developmental process and stress tolerance, genes involved in biosynthesis processes (glycosylt), and stress-responsive proteins such as protein phosphatase, K+ ion channel, and receptor-like kinase, demonstrated significant associations with the phenotypic trait. These findings underscore the efficiency of Et-GWAS in identifying variants and loci associated

with the target trait. Notably, the gene *OsPEX1* displayed two top-performing haplotypes in different background genomes (Fig 5A–D and Table S8). Both haplotypes exhibited high performance for grain yield under drought stress, emphasizing the role of epistatic interactions between the gene and its surrounding regions in the overall performance of the variety.

We investigated the non-synonymous SNPs in these superior haplotypes and the predicted structural variations between the reference and superior haplotypes, as presented in Table S8 and Figs S6 and S7A and B. Based on the top-performing haplotypes, we have identified 13 accessions as potential donors for grain yield under drought stress in resilient breeding programs. These accessions consistently demonstrated high performance across both seasons, despite experiencing varying levels of stress conditions. The accessions possessing these superior haplotypes are listed in Table S9, and notably, all these accessions have a DSI of less than one in both seasons.

## Validation of Et-GWAS results

Et-GWAS successfully identified several MTAs within the regions of previously reported QTLs for grain yield under drought-stress conditions. Specifically, 24 MTAs were detected (22 from DS2019 and two from DS2020) within the reported QTL *qDTY 2.2* range, with a PVE ranging from 3.6% to 4.6%. Using a window size of 100KB, six MTAs overlapped with *qDTY 1.1* and *qDTY 12.1*. In addition to these direct QTLs, some MTAs were found within 200KB of two root QTLs, *qPRDW-12-1*, and *qPRT-12-1*, related to drought tolerance, root thickness, and weight. In addition, one MTA was within 50KB of a yield and productivity QTL, *qYLD-2-1*. These findings highlight the consistent association of MTAs identified by Et-GWAS with known QTLs for drought tolerance and productivity, supporting the

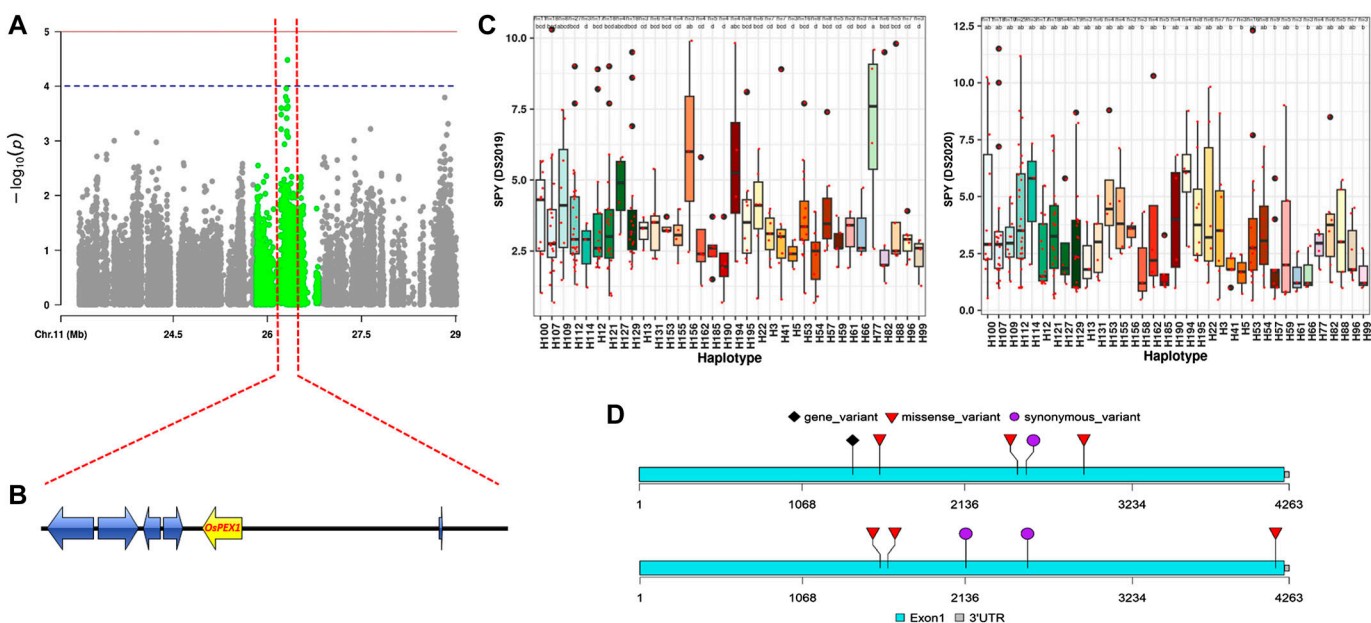

**Figure 5. Haplotype analysis of the *OsPEX1*.**
**(A)** The 100-kb window (centered on the MTA) length chosen to identify the candidate genes. **(B)** Six genes were identified (with LD> = 0.4) for the further haplo-pheno analysis. **(C)** Shows the SPY variations among the haplotypes from DS2019 & DS2020. **(D)** Two top-performing haplotypes of *OsPEX1* with their constituting mutation positions are shown here. The performance of the haplotypes depends on the genomic background.

effectiveness of this approach in revealing genetic factors underlying grain yield under drought stress conditions in rice. Detailed information regarding the MTAs overlapping with known QTLs is provided in Table S10.

We conducted a comparison between the MTAs identified through Et-GWAS and conventional GWAS methods. Et-GWAS considered the bulks as linear points, whereas the GWAS used the mixed linear model (MLM) for analysis. For the DS2019 data, GWAS using MLM identified 56 MTAs (Table S11), with 19 located within reported QTLs (*qDTY 1.2* and *qDTY 2.2*). Similarly, for the DS2020 data, GWAS associated 26 SNPs with grain yield under drought stress, with two found within the reported QTL region (*qDTY 2.2*) (Table S11). The PVE of the identified MTAs ranged from 4.1% to 6%, and the allele effect ranged from −1.3 to 1.5 (Fig S8A and B). With a 100KB bin size for the comparative study between GWAS and Et-GWAS, we observed 18 overlapping MTAs from the DS2019 data and two overlapping MTAs from the DS2020 data, resulting in a ~35% overlap between the studies (20 out of 56). These results indicated that Et-GWAS was able to detect association signals that conventional GWAS might have missed. Comparing the MTAs identified from both seasons through window bins (100KB bin size) revealed overlapping regions on chromosome 2.

To validate the effectiveness of Et-GWAS, we focused on identifying MTAs associated with resistant starch (a sub-trait of the glycemic index) and compared the results with a published GWAS study. Using a bulk size of 15, we successfully identified regions that overlapped with six out of the nine reported MTAs. Within each of these regions, multiple MTAs were detected, resulting in a ~66% overlap between the MTAs identified by Et-GWAS and conventional GWAS approaches. Fig S9 illustrates the results obtained with a bulk size of 10, wherein the identified region on the sixth chromosome

showed multiple MTAs under the Et-GWAS approach. This finding indicates that Et-GWAS efficiently concentrates low-frequency variants through extreme sampling, enabling the detection of multiple MTAs within a specific genomic region.

## Discussion

Significant progress has been made in rice research since the first high-quality rice genome was completed in 2005 (Sasaki, 2005). The 3,000 Rice Genomes Project (3K RG) further contributed to this progress by providing publicly available genome sequences of 3,010 rice accessions, representing global genetic and functional diversity, thus fuelling research in rice functional genomics (Hirochika et al, 2004; Zhang et al, 2008). However, despite these valuable resources, the application of rice genetics and genomics research findings has not yet led to fundamental changes in rice breeding practices. The publicly available sequence data remained largely untapped by most rice breeders and the global community in their breeding programs. The underutilization of publicly available sequence data in rice breeding practices is primarily attributed to the gap between breeders, who possess profound crop understanding, and data scientists, who have coding proficiency. As a result, the full potential of 3K RG information has not been fully harnessed. To address this issue and bridge the gap between breeders and data scientists, it is essential to develop analysis tools that can be easily used by the non-data scientist community. Enabling breeders to use their extensive knowledge in decision-making tools will unlock the true potential of the 3K RG data and lead to more effective rice breeding strategies.

The recent progress in genomics within crop science has provided an abundance of genomic sequence data, high-throughput phenotypic information under diverse conditions, and a variety of statistical tools to analyze these data. Among these techniques, GWAS has emerged as a prominent method in the post-genomic era, despite its limitations, for identifying markers associated with agronomically important traits (Huang et al, 2010; 2011; Zhao et al, 2011; Morris et al, 2013; Yang et al, 2014; Yano et al, 2019; Maldonado et al, 2019). Our study focused on addressing missing heritability by incorporating low to medium-effect SNPs and reducing sequencing costs through the use of current genomic sequence information. We adopted a pooling strategy from the extreme-phenotype GWAS (Yang et al, 2015) and evaluated its efficiency in rice to identify MTAs associated with grain yield under drought-stress conditions. Various alternative techniques and improvements have been implemented in association studies to reduce the number of association markers and overcome missing heritability.

Grain yield is a complex trait influenced by multiple genes, and its heritability under drought is moderate (Kumar et al, 2007), making it a crucial selection criterion for drought-resilient crop breeding. Previous studies have reported several large-effect QTLs for grain yield under drought, individually explaining 10–30% of the trait heritability (Lafitte et al, 2004; Yue et al, 2006). Notably, some of these QTLs, such as *qDTY 12.1* and *qDTY 3.1*, explain a higher percentage of genetic variation (51% and 36%, respectively) (Bernier et al, 2007; Dixit et al, 2015). In our study, Et-GWAS identified 54 signals associated with grain yield under drought in rice during the wet seasons of 2019 and 2020, with significant SNPs present on all chromosomes except chromosome 7. The comparative analysis demonstrated the accuracy and resolution of Et-GWAS, as it successfully identified SNPs within the reported large-effect QTLs for yield under drought. Furthermore, our method showed significant overlapping SNPs with the results of GWAS MLM, highlighting the precision of Et-GWAS, where extreme phenotypic bulks served as efficient alternatives to extensive genetic panels, as anticipated.

Et-GWAS not only identified overlapping regions but also revealed 30 additional signals associated with yield under drought, which were not identified through MLM. Among these, 15 MTAs are located within genic regions, and functional characterization of the genes in this list suggests their involvement in stress response pathways. For instance, the *OsNPC2* gene codes for phospholipase, playing a role in plant growth by strengthening the stem nodes (Cao et al, 2016). Another gene, *LAZY2* (*OsLA2*), is known to control the tiller angle (Huang et al, 2021), whereas *OsPEX1* affects plant growth through lignin biosynthesis (Ke et al, 2019). We identified 141 candidate genes from both seasons (DS2019 and DS2020) that are in high linkage with significant SNPs. Haplo-pheno analysis of these candidate genes revealed 12 genes (6 from DS2019, 2 from DS2020, and 4 from both seasons) with top-performing haplotypes in our diversity panel.

Drought stress during the reproductive stage of rice significantly affects its development and grain formation (Hargurdeep & Saini, 1999). We identified a superior haplotype of the gene *Os01g0755100* (*OsAMD1/OsPWA1*), known to exhibit a defective pollen wall structure and cause male sterility, which is associated with improved yield under drought stress conditions (Zhang et al, 2022). Another significant finding is the haplotype of the protein

phosphatase gene *Os02g0180000* (*OsPP2C11*), which possesses a tolerated nonsynonymous mutation and is linked to yield under drought stress. In addition, the gene *OsK5.2*, encoding an outward shaker K+ channel, plays a vital role in regulating stomatal closure and K+ transport in response to drought stress (Nguyen et al, 2017). Our method also identified the basic helix-loop-helix (bHLH) gene haplotype *Os01g0952800* (*OsIRO2/OsbHLH056*) associated with drought tolerance, a gene family known for its involvement in abiotic stress responses (Waseem et al, 2019). Furthermore, we identified the Pumilio/Fem-3 binding factors gene haplotype *Os12g0488900* (*Ospuf4*), which acts as a negative regulator under drought, osmotic, and salt stress conditions (Huh & Paek, 2014).

We have discovered a metal-dependent hydrolase gene (*Os01g0952700*) haplotype associated with high yield under drought stress conditions. This haplotype contains two missense mutations (M249I and S489F) predicted to affect the α-helixes, with SIFT scores of 0.089 and 0.077, respectively. Interestingly, the superior haplotype of this gene was found in four known drought-tolerant varieties (Shabhagi Dhan, DRR Dhan 44 and 46, IR 128773-4-3-1-4-B, and Vandana) but was absent in all susceptible check varieties (BPT 5204, Naveen, Co-51, DRR Dhan 48) (Fig S7A and B). These results highlight the effectiveness of the Et-GWAS approach in selecting MTAs and haplotypes for targeted breeding strategies.

Et-GWAS results overlap in the range of 30–60% with conventional GWAS depending on the study and variant dataset. Based on the bulk sampling and the analysis results from our study, we draw the following conclusions: the Et-GWAS yields overlapping results with GWAS that are identified based on the common variants. Et-GWAS also identifies novel MTAs that are either diluted or overlooked in the standard GWAS analysis. These MTAs are identified by focusing on the concentration of rare variants found in the extreme phenotypic performing lines.

To make the Et-GWAS more accessible to the researchers we have also created a user-friendly graphical interface for Et-GWAS using ShinyApp, which can extract sequence information from the Open Source based on the genotype list. Our study demonstrates several key advantages of this application.

(1) Efficient in identifying rare alleles with low to minimal impact. Thus, it increases the true negatives in the association study.
(2) It uses open source to download the sequence data. The pipeline uses the SNP-seek database (https://snp-seek.irri.org/) of the 3k rice genome project. Moreover, it can be modified to any other open-source database.
(3) It is made as a streamlined pipeline that identifies association signals (MTAs) and proceeds further to the identification of superior haplotypes associated with the trait. Because it was made as a GUI that can be run in server (for low coverage) and locally (for high coverage) it will be easy to use for the non-computational research community.
(4) The application can be run from a server and and locally using in RStudio. Thus it gives the flexibility for the users to explore with various parameters and low- and high-density marker datasets.

Our innovative Et-GWAS method, combining extreme bulk pooling and allele frequency measurement, efficiently uncovered

associations between genetic variants and grain yield under drought stress conditions. The developed architecture showcases remarkable versatility, enabling seamless adaptation for trait mapping studies in diverse crops and species. This opens up exciting opportunities to accelerate crop improvement beyond rice. In addition, the GUI enhances the usability of Et-GWAS for researchers in the field of crop science, further facilitating its widespread application and impact.

# Materials and Methods

### GUI implementation (app development)

The Et-GWAS shiny app is entirely written in the R language and bash scripts, with the underlying R code encapsulated by the shiny R package (Chang et al, 2023), which is a web application framework for R, offering an interactive GUI. Et-GWAS is hosted by a Shiny web server (https://et-gwas.shinyapps.io/Et-GWAS/), where the association is studied with 30K SNP dataset. The dataset was downloaded from The International Rice Informatics Consortium's (IRIC) SNP-seek database (https://snp-seek.irri.org/_download.zul) (Alexandrov et al, 2015; Mansueto et al, 2017; Wang et al, 2018) and filtered for minor allele frequency above 0.1 and pair-wise LD above 0.3 (commands used in plink: −maf 0.1 −indep-pairwise 50 10 0.3). The application can also be run locally within RStudio by running the source code from the GitHub repository (https://github.com/IRRI-South-Asia-Hub/Et-GWAS). From the downloaded directory, the source file named app.R in RStudio can be run by clicking the *Run App* button. The details to use this source code are given with steps in the documentation present at GitHub. When the Et-GWAS shiny app run in a dual processor workstation with 128 GB RAM and Intel Xeon Silver 4110 2.1 GHz (3.0 GHz Turbo, 8C, 9.6 GT/s 2UPI, 11 MB Cache, HT [85W] DDR4-2400), the elapsed time was 4,844 s (11,092 s user CPU time) to finish the analysis with high density SNP dataset.

### Plant materials and drought-tolerance phenotyping

A panel of 399 early and medium maturing lines was created as a subset of 3,000 accessions of the Rice Genome Project (Li et al, 2014). These lines were subjected to reproduction stage (RS) drought stress, as explained previously (Kumar et al, 2014) in the 2019 and 2020 dry seasons. A non-stress experiment was also maintained alongside for control. Matured panicles were collected, and the corresponding SPY in grams was calculated for each accession in three replicas.

### Genomic data quality control and population structure

The genomic data for the 399 accessions were downloaded from IRIC's SNP-seek database. The LD pruned GWAS SNP set was used, which was filtered ($r^2$ cutoff = 0.8, window size = 2 kb, step size = 1 SNP) from the 4.8 M SNP dataset. This dataset contains 1011601 SNPs, and it was further filtered using PLINK 2.0 (Purcell et al, 2007) to prepare the working dataset. Two variations of filtering options were checked to optimize the quality control of the genomic data for Et-GWAS; (a) stringent filtering: removal of SNPs with MAF less than 0.05 (--maf 0.05 −

removing SNPs with MAF less than or equal to 0.05) and missingness lesser than 0.1 (--geno 0.1; removing SNPs with >10% missing genotypes), and removal of individuals with missingness (--mind 0.1; removing genotypes with >10% missing SNPs) lesser than 0.1. (b) Minimum filtering: removal of SNPs with MAF less than 0.05 and sequence depth equivalent to the bulk size. The stringent filters resulted in 518820 SNPs and the minimum filtering resulted in 600288 SNPs which retained important SNPs; hence the minimum filtering was followed for the further analysis in Et-GWAS. For the rest of pre-GWAS analysis and conventional GWAS, the slightly low-density SNP dataset (with 518820 SNPs) was used. The frequency summary and the quality control of the genomic data were carried out using PLINK 2.0 along with the principal component analysis of this panel. The first three principal components were plotted using a custom R script. The genetic diversity was studied using the Neighbour-joining clustering method using TASSEL 5.0 (Bradbury et al, 2007). The unrooted representation of this phylogenetic tree was constructed using iTol (Letunic & Bork, 2021). The population genetic structure of the panel was estimated based on the maximum likelihood from the SNP dataset using the ADMIXTURE 1.3.0 (Alexander et al, 2009; Alexander & Lange, 2011). Analysis was carried out for a range of K values (K = 1 to K = 6), and the three significant subpopulations were visualized with K = 3 using R script.

### Bulk sampling and measurement of pooled allele frequency

The percentage of accessions to be included in the bulks is user-defined. Based on the percentage, the accessions with extreme phenotypic values are grouped as low and high bulks, with the same size a bulk with randomly chosen accessions that will have mean and median trait values like the diversity panel. The alleles of each SNP are then clubbed for each bulk, and their corresponding frequency is calculated as a reference and alternate allele frequency. As reported earlier, the bulk size depends on homozygosity, segregation, and the sequence depth (Magwene et al, 2011; Tiwari et al, 2016). Hence, we have compared various percentages of bulk sizes (10, 15, and 20% of samples from the diversity panel) to finalize the optimum pool size, which is 20%. The subset generated through the bulk formation shows complex population stratification (Fig S10A–D). The selected SNPs are then analyzed individually by considering the low, random, and high bulks as linear sequences through the GLM. To account for population stratification that was observed, we have included an inflation factor to the *P*-value computation as a precaution (Devlin & Roeder, 1999). The SNPs that show significant monotonous trends will be considered markers associated with the trait of interest. The sequencing depth is calculated from the reference allele frequency ($ref_{af}$) from three bulks as follows,

$$Total\ depth = \left(2 \times Low_{ref\_af}\right) + \left(2 \times High_{ref\_af}\right) + \left(2 \times Rand_{ref\_af}\right)$$

### Association analysis

The whole dataset was analyzed for comparative analysis by considering fixed and random effects through four different statistical methods (GLM, MLM, FarmCPU, Blink). These analyses were

done in GAPIT (Lipka et al, 2012), where the tool calculated principal components and kinship matrices. The dataset with 518820 SNPs using the stringent filtering method was used for this analysis. Among those methods, MLM produced results closer to the Et-GWAS, hence, only MLM was results were considered for the comparison.

The Et-GWAS is carried out using the GLM package in R, and the script from Jinliang Yang's GitHub source code was modified according to our need (Yang et al, 2015). The analysis will finally result in the *P*-value, $R^2$ value, and PVE for each SNP. Genomic datasets from both filtering options were considered with three different bulk sizes.

$$PVE = \frac{2\beta^2(MAF)(1-MAF)}{2\beta^2(MAF)(1-MAF) + 2 \times se(\beta)^2 N(MAF)(1-MAF)}$$

### Overlapping MTAs

Unambiguous replication of results from various association studies is unlikely because most traits' genetic architecture is complex in nature (Johnson et al, 2011). Hence, to compare the results from different season datasets and methodologies, we need to follow the overlapping MTAs within the LD region. We did this by breaking the genome into bin sizes of 100 kb, and the MTAs from various datasets/methods that fall in the same bin are considered consistent MTAs. We term them as overlapping MTAs.

### Candidate genes identification

Association studies are done using moderately dense SNP datasets, so getting MTAs in the genic region is rare. We can identify the plausibly associated genes with our trait of interest by considering the linkage. SNPs present within the 100-kb interval centered on the detected MTAs were considered for the identification of the candidate genes. The SNPs with high linkage ($r^2 \geq 0.4$) were extracted from this. The genes harboring these SNPs were identified using the Nipponbare reference annotation file (gff; Os-Nipponbare-Reference-IRGSP-1.0) from the Rice Annotation Project Database (rap-db) at https://rapdb.dna.affrc.go.jp/. These genes are further explored in the rap-db for their functional characterization (Sakai et al, 2013). These genes are called candidate genes.

### *Haplo-pheno* analysis

The variants (all the SNPs) present in the candidate genes for the whole 3,000 accessions were obtained from SNP-seek database (Mansueto et al, 2017) using an in-house automation script written in Python using the selenium package. The script is available at the Python Package Index (https://pypi.org/project/Sudip-snpdw/) to download. The information regarding the installation and usage are given in the GitHub page of Et-GWAS (https://github.com/IRRI-South-Asia-Hub/Et-GWAS). These variants are converted into aligned sequences of DNA, and the identical DNA sequences are collapsed into haplotypes using the haplotype R package (Aktas, 2020) (v1.1.2). Whereas identifying the haplotypes in the 3k rice population, we included the entire variation; even the haplotype possessed by single accession was identified. From the initial

haplotypes, we only considered the haplotypes possessed by two or more accessions. The phenotypic performance of these haplotypes is then used to organize them into significantly different groups based on Duncan's test using the agricolae R package (de Mendiburu, 2021) (v1.3-5). Top-performing haplotype groups are identified when their average phenotypic performance is higher than one above the mean phenotypic performance of the diversity panel.

### Validation of developed pipeline using resistance starch trait

The resistant starch (RS) of rice is the most critical factor responsible for breeding low glycemix index (GI). A recent study (Selvaraj et al, 2021) published a set of MTAs associated with rice GI and other nutritional quality traits. The resistance starch values of the panel containing 186 lines from the study were used for the analysis. The filtered set from 3K 1M GWAS SNP dataset was used for the association study. Though the number of SNPs used by us and this study was almost equal, it is not the exact same genotypic dataset. And hence, the results from our study were compared by keeping 1 MB as window size. Three different bulksizes were used to identify the RS-associated variants.

### Validation of superior haplotypes by comparing with tolerant genotypes

The variant calling files of tolerant and susceptible genotypes, along with the superior haplotypes of the candidate genes, are merged and annotated using Vcftools (v0.1.13) (Danecek et al, 2011) and SNPEff (v5.1) (Cingolani et al, 2012), respectively. Aliview v1.28 (Larsson, 2014) was used to visualize the variants among the compared genotypes. The 3D protein structure was predicted by SWISS-MODEL (Waterhouse et al, 2018).

## Data Availability

The whole-genome sequencing data from this publication has been deposited to the National Center for Biotechnology Information (NCBI) under accession number PRJNA1055582. The source code of the Et-GWAS shinyapp is freely available at https://github.com/IRRI-South-Asia-Hub/Et-GWAS. The documentation file is also there to provide the details regarding installation and usage of the R package locally and as web tool. The bash scripts and R functions needed to carry out important pre-association analysis such as quality control of genotypic data (using PLINK) and visualization of population stratification (PCA and k-means cluster analysis using ADMIXTURE) are also present in the same GitHub under the scripts folder.

## Supplementary Information

# Acknowledgements

The authors would like to thank the Department of Biotechnology (DBT), the Government of India, and the DBT-RA program in Biotechnology and Life Sciences. This work has been undertaken as part of the ICAR-IRRI collaborative research project. IRRI is a member of the CGIAR Consortium.

## Author Contributions

N Gnanapragasam: resources, data curation, software, formal analysis, validation, investigation, visualization, methodology, and writing—original draft, review, and editing.

VV Prasanth: resources, data curation, methodology, and data generation.

KT Sundaram: resources and software.

A Kumar: resources and software.

B Pahi: resources and data curation.

A Gurjar: resources and data curation.

C Venkateshwarlu: resources, data curation, and data generation.

S Kalia: funding acquisition and writing—review and editing.

A Kumar: supervision, funding acquisition, and writing—review and editing.

S Dixit: writing—review and editing.

A Kohli: supervision and writing—review and editing.

UM Singh: supervision, investigation, project administration, and writing—review and editing.

VK Singh: conceptualization, supervision, funding acquisition, investigation, project administration, and writing—original draft, review, and editing.

P Sinha: conceptualization, supervision, funding acquisition, investigation, project administration, and writing—original draft, review, and editing.

## Conflict of Interest Statement

The authors declare that they have no conflict of interest.

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
