## [Reviewer comments · Life Science Alliance]

Life Science Alliance

Extreme trait GWAS (Et-GWAS): Unraveling rare variants in the 3000 rice genome.

Niranjani Gnanapragasam, Vinukonda Prasanth, Krishna Sundaram, Ajay Kumar, Bandana Pahi, Anoop Gurjar, Challa Venkateshwarlu, Sanjay Kalia, Arvind Kumar, Shalabh Dixit, Ajay Kohli, Uma Singh, Vikas Singh, and Pallavi Sinha

DOI: <https://doi.org/10.26508/lsa.202302352>

Corresponding author(s): Pallavi Sinha, International Rice Research Institute and Vikas Singh, International Rice Research Institute (IRRI), South Asia Hub

Review Timeline:

Submission Date:	2023-09-01
Editorial Decision:	2023-10-16
Revision Received:	2023-11-14
Editorial Decision:	2023-12-05
Revision Received:	2023-12-12
Accepted:	2023-12-13

Transaction Report:

October 16, 2023

Re: Life Science Alliance manuscript #LSA-2023-02352-T

Pallavi Sinha
International Rice Research Institute (IRRI), South Asia Hub

Dear Dr. Sinha,

Thank you for submitting your manuscript entitled "Extreme trait GWAS (Et-GWAS): Unraveling rare variants in the 3000 rice genome." to Life Science Alliance. The manuscript was assessed by expert reviewers, whose comments are appended to this letter. We invite you to submit a revised manuscript addressing the Reviewer comments.

Thank you for this interesting contribution to Life Science Alliance. We are looking forward to receiving your revised manuscript.

Sincerely,

B. MANUSCRIPT ORGANIZATION AND FORMATTING:

Reviewer #1 (Comments to the Authors (Required)):

The article entitled 'Extreme trait GWAS (Et-GWAS): Unraveling rare variants in the 3000-rice genome' used Extreme trait GWAS for identification of rare alleles. This is a great advancement in conventional GWAS analysis, initially used by Yang et al. 2015. I enjoyed reading the manuscript and come up with friendly points for discussion and raised some of the issues for further improvement of the manuscript.

1. What is the difference between XP-GWAS (Yang et al. 2015) (<https://onlinelibrary.wiley.com/doi/pdf/10.1111/tpj.13029>) and Et-GWAS in this study.
In abstract
2. If the Et-GWAS provided the similar results as conventional GWAS then what is the significance of Et-GWAS--- This is a great point of discussion.
3. How the developed online GUI tool will be helpful for the community not accessible to 3000 rice genotypes. Just a doubt beyond the objectives and findings of this study.
4. Why the SNPs were analyzed using GLM? Did you try population structure among low, high and random pools?
5. What if you used BLUPs calculated from two seasons for GWAS analysis? Or combined analysis of the data from two seasons.
6. At one place it is mentioned as missingness is not used for filtration of SNPs, however, in material and methods section it is mentioned as missingness threshold of 0.01. Please check, as there is difference between no threshold and a threshold of 0.01 and in population structure it is mentioned as 10% missing were used.
7. In results it is mentioned as GLM is used for GWAS (page 8 last para) and in figure 1 as well. However, in material and methods it is mentioned as GLM, MLM, and FarmCPU and Blink from GAPIT. Please check, if I am not wrong.
8. It would be great if the authors could make available the haplo-pheno analysis python package selenium.
9. At the end of page 4 and in the beginning of page 5. The limitations mentioned for XP-GWAS are truly unconvincing. Making pools is not a challenge, so I don't think that it can be a limitation. Also, it is mentioned that there are two limitations and I don't see clearly two limitations in the text. I think this para needs to be revised completely.
10. graphical interface (GUI). Here the 'user' word is missed on page 5 second para
11. Page no 26. The panel of 399 lines is developed based on maturity of 3K lines. How we can say that the Pools for yield per plant are developed based on 3K rice accessions? You must have missed extreme trait representatives from 3K panel, as you don't have accessions for extremely high and low per plant yield from 3K. It would have been convincing if the 399 panel would have been developed based on per plant yield from 3K rice genomes.
12. The R script for determining $K=3$ should be provided in Supplementary for readers.

Reviewer #2 (Comments to the Authors (Required)):

My feeling is that the paper is largely solid scientifically but not outstandingly novel. I fear that the invention of ever more fancy GWAS based acronyms is in itself not helpful and that the approach of using stress conditions to access a fuller coverage of the genome has already been postulated and even demonstrated by the Fernie group in Golm. As such I feel that the most novel aspect is the "Unraveling of rare variants in 3000 rice genomes" however to my opinion whilst the title is highly enticing I am not fully convinced that they actually achieved this. I would thus suggest that they refocused the ms to this point and provided stronger evidence or alternative submitted this work to a more specialized journal.

Author's Response to the Reviewer's

We would like to express our gratitude to the Handling Editor for providing valuable feedback on our manuscript titled "**Extreme trait GWAS (Et-GWAS): Unraveling rare variants in the 3000 rice genome**" and accepting the MS for review. We appreciate the thorough evaluation and insightful comments that have greatly helped us to improve our manuscript. Based on the Reviewer's suggestions, we have carefully revised the manuscript. We have addressed each comment point by point, incorporating necessary changes, and clarifications where applicable. We have made significant revisions to the results section with a detailed explanation, and provided more in-depth discussions on the implications of our findings.

We believe that these revisions have substantially improved the manuscript and have enhanced its scientific rigor and clarity. We are confident that the revised version will meet the standards and expectations of the journal.

Reviewer #1

Reviewer Comment 1: What is the difference between XP-GWAS (Yang et al. 2015) (<https://onlinelibrary.wiley.com/doi/pdf/10.1111/tbj.13029>) and Et-GWAS in this study.

Authors' Response: The authors are thankful to the Reviewer for the observation and thorough examination of both methods of trait mapping.

Following are the major differences between the XP-GWAS and Et-GWAS:

- Since we are making use of the existing 3K RG sequence datasets that are currently accessible, there is no need to generate a pooled sample genome sequence. The user-defined pool size and phenotypic characteristic values are used by the pipeline to construct the pooled allele frequency.
- Additionally, our study encompasses the preparation of pre-GWAS data and the identification of candidate genes based on the GWAS analysis.

Reviewer Comment 2: If the Et-GWAS provided the similar results as conventional GWAS then what is the significance of Et-GWAS--- This is a great point of discussion.

Authors' Response: Et-GWAS combines extreme bulk pooling and allele frequency measurement to facilitate the detection of associations between genetic variants and target traits. This method effectively concentrates and quantifies rare alleles by leveraging the available genome sequence information. In the case of conventional GWAS, the variants identified through it can only explain a

certain level of heritability of the targeted trait. Furthermore, GWAS uses stringent thresholds in data pre-processing (filtering) and association analysis (multiple-test correction, Bonferroni correction) to reduce false positives which causes the elimination of rare alleles.

Reasoning to show that Et-GWAS can identify novel MTAs has been added in the Discussion part on page no. 20, 2nd paragraph:

“Et-GWAS results overlaps in the range of 30-60% with conventional GWAS depending on the study and variant dataset. Based on the bulk sampling and the analysis results from our study, we draw the following conclusions: The Et-GWAS yields overlapping results with GWAS that are identified based on the common variants. Et-GWAS also identifies novel marker trait associations (MTAs) that are either diluted or overlooked in the standard GWAS analysis. These MTAs are identified by focusing on the concentration of rare variants found in the extreme phenotypic performing lines”.

Reviewer Comment 3: How the developed online GUI tool will be helpful for the community not accessible to 3000 rice genotypes. Just a doubt beyond the objectives and findings of this study.

Authors’ Response: Thank you for this very useful comment. The Shiny app for Et-GWAS is accessible through two modes:

1. Web Server: Users can access this mode through a web interface, enabling them to run the analysis with simple clicks and file uploads. This mode utilizes a genotypic dataset containing approximately 30,000 SNPs. User doesn’t required to download any genotypic dataset to use the web server.

2. R-based ShinyApp: Users can choose this mode if they wish to install the package by following the steps outlined in the documentation. This mode works with a larger genotypic dataset comprising 600288 SNPs, which have been filtered. This dataset has been uploaded in the Github and it will be downloaded along with the shiny app installation in the R.

Both datasets are sourced from the 3K-RG website. If a user wants to customize the genotypic dataset further, for example, by including only functional SNPs or INDELs, or by applying other custom filtration criteria, they have the option to directly modify the code available on the GitHub page. Detailed instructions on how to make these modifications are provided in the app documentation uploaded on the same GitHub page (<https://github.com/IRRI-South-Asia-Hub/Et-GWAS/>). The page included:

- Documentation to include the information regarding the downloading of data and how to use it.
- Documentation for modifying the code, to use a different set of SNP dataset.

Reviewer Comment 4: Why the SNPs were analysed using GLM? Did you try population structure among low, high, and random pools?

Authors' Response: GLM: The primary goal of the pooling strategy in Et-GWAS is to preserve rare alleles that tend to become diluted when analysing the entire genetic panel. This dilution issue is also encountered when employing complex statistical models such as mixed or multi-locus models to identify the associations. Therefore, we opted for the simplest model to identify the associations between genetic variants and the trait.

Population stratification removal at pooling: At each pooling step, we carefully assess the distribution of the rice subpopulations in the high, low, and random pools. During the initial optimization process, we constructed the pools with consideration of these subpopulations. If any of the pools contain a subpopulation that is not represented in the other two pools, we removed that specific subpopulation from the panel, and the pooling process was restarted. The effectiveness of this strategy depends on the composition of the panel.

Based on our ongoing research, particularly with submergence stress trials, we hypothesize that this pooling strategy (by bulking the samples based on the phenotypic performance and the population structure) will perform effectively when dealing with stress-scoring data, which is quantitative in nature.

Population stratification in Et-GWAS: Having said that, we prepared bulks by only considering the phenotypic trait value in our current study. We noticed a complex population stratification due to the reduction in sample size from its original panel. The corresponding distribution of principal component analysis plot has been included in the supplementary Figure (Figure S10). On the account of this potential population stratification, the Et-GWAS analysis calculated the inflation factor for the bulk subset during the P-value computation (Devlin and Roeder, 1999). This information was added in the Materials and Methods (“Bulk sampling and measurement of pooled allele frequency”) section on page no. 25, 2nd paragraph:

“The percentage of accessions to be included in the bulks is user-defined. Based on the percentage, the accessions with extreme phenotypic values are grouped as low and high bulks, with the same size a bulk with randomly chosen accessions that will have mean and median trait values like the diversity panel. The alleles of each SNP are then clubbed for each bulk, and their corresponding frequency is calculated as a reference and alternate allele frequency. As reported earlier, the bulk size depends on homozygosity, segregation, and the sequence depth (Magwene et al., 2011; Tiwari et al., 2016). Hence, we have compared various percentages of bulk sizes (10,15 & 20% of samples from the diversity panel) to finalize the optimum pool size, which is 20%. The subset generated through the bulk formation shows a complex population stratification (Figure S10). The selected SNPs are then analyzed individually by considering the low, random, and high bulks as linear sequences through the

general linear method (GLM). To account for population stratification that was observed, we have included an inflation factor to the P-value computation as a precaution (Devlin and Roeder 1999). The population stratification. The SNPs that show significant monotonous trends will be considered markers associated with the trait of interest. The sequencing depth is calculated from the reference allele frequency (ref_{af}) from three bulks as follows”,

$$Total\ depth = (2 * Low_{ref_af}) + (2 * High_{ref_af}) + (2 * Rand_{ref_af})$$

Reviewer Comment 5: What if you used BLUPs calculated from two seasons for GWAS analysis? Or combined analysis of the data from two seasons.

Authors’ Response: Ideal scenario would be considering the combined phenotypic data from both seasons. However, due to the unique circumstances surrounding this study (Pandemic outbreak happened in the midst of the first season), notably the stress conditions and the differences between the two seasons, we found that the imposed stress significantly varied. This discrepancy was evident when we analysed the Pearson correlation of the Drought Susceptibility Index (DSI) and conducted the Kolmogorov-Smirnov test (Figure_S3) on the data from the two seasons. As a result of these significant variations, we made the decision to treat the two seasons as distinct environments and keep their data separate.

Reviewer Comment 6: At one place it is mentioned as missingness is not used for filtration of SNPs, however, in material and methods section it is mentioned as missingness threshold of 0.01. Please check, as there is difference between no threshold and a threshold of 0.01 and in population structure it is mentioned as 10% missing were used.

Authors’ Response: The analysis was tried with multiple filtering options to create a more manageable working SNP dataset. Two major filtering options were used, where the stringent filtering (based on minor allele frequency and missingness) yielded 518820 SNPs. This dataset was utilized in the population structure analysis and conventional GWAS. Whereas the minimum filtering option (based on minor allele frequency alone) yielded in 600288 SNPs, which was used for the Et-GWAS analysis. The confusion in expressing this information was noted and rectified in the methods section.

The details of the SNP data filtering are modified to explain the two sets of filtering used in the analysis under Methods and Materials (Genomic data quality control and Population structure) on page no. 24, 2nd paragraph:

The genomic data for the 399 accessions were downloaded from IRIC’s SNP-seek database. The LD pruned GWAS SNP set was used, which was filtered (r^2 cutoff = 0.8, window size = 2kb, step size = 1

SNP) from the 4.8 M SNPs dataset. This dataset contains 1011601 SNPs, and it was further filtered using PLINK 2.0 (Purcell et al. 2007) to prepare the working dataset. Two variations of filtering options were checked to optimize the quality control of the genomic data for Et-GWAS; (a) stringent filtering: removal of SNPs with MAF less than 0.05 (--maf 0.05 – removing SNPs with MAF less than or equal to 0.05) and missingness lesser than 0.1 (--geno 0.1 – removing SNPs with >10% missing genotypes), and removal of individuals with missingness (--mind 0.1 – removing genotypes with >10% missing SNPs) lesser than 0.1. (b) minimum filtering: removal of SNPs with MAF less than 0.05 and sequence depth equivalent to the bulk size. The stringent filters resulted in 518820 SNPs and the minimum filtering resulted in 600288 SNPs which retained important SNPs, hence the minimum filtering was followed for the further analysis in Et-GWAS. For the rest of pre-GWAS analysis and conventional GWAS, the slightly low density SNP dataset (with 518820 SNPs) was used.

Reviewer Comment 7: In results it is mentioned as GLM is used for GWAS (page 8 last para) and in figure 1 as well. However, in material and methods it is mentioned as GLM, MLM, and FarmCPU and Blink from GAPIT. Please check, if I am not wrong.

Authors' Response: While we acknowledge that the use of complex statistical models like mixed or multi-locus models can potentially lead to the loss of rare variants, during the initial optimization phase, we explored multiple methods for comparison with the results obtained from the Extreme Trait GWAS (Et-GWAS). Based on these findings, we chose to incorporate only the results from the Mixed Linear Model (MLM) in the final analysis. This specific information has been rectified in the Materials and Methods section (Association analysis) on page no. 26, 1st paragraph:

“The whole data set was analyzed for comparative analysis by considering fixed and random effects through four different statistical methods (GLM, MLM, FarmCPU, Blink). These analyses were done in GAPIT (Lipka et al. 2012), where the tool calculated principal components and kinship matrices. The dataset with 518820 SNPs using the stringent filtering method was used for this analysis. Among those methods MLM produced results closer to the Et-GWAS hence, only MLM was results were considered for the comparison”.

Reviewer Comment 8: It would be great if the authors could make available the haplo-pheno analysis python package selenium.

Authors' Response: The script is available at the Python Package Index (<https://pypi.org/project/Sudip-snpdw/>) to download. The information regarding the installation and

usage has been added in the github page of Et-GWAS (<https://github.com/IRRI-South-Asia-Hub/Et-GWAS.git>)

This information is added in the Materials and Methods (Haplo-Pheno analysis) on page no. 27, 2nd paragraph:

Reviewer Comment 9: At the end of page 4 and in the beginning of page 5. The limitations mentioned for XP-GWAS are truly unconvincing. Making pools is not a challenge, so I don't think that it can be a limitation. Also, it is mentioned that there are two limitations and I don't see clearly two limitations in the text. I think this para needs to be revised completely.

Authors' Response: The section has been modified to explain the limitation regarding the pooling of sequence information from the existing sequence data. And the error regarding the number of limitations has been rectified and can be read on page no. 4 last paragraph and page no. 5 first paragraph:

“Pooled-based Genome-Wide Association Studies (GWAS) utilizing extreme phenotypes have been applied in both plant and human genetics, with various techniques such as Pool GWAS, XP-GWAS, and Pooled sample-based GWAS. However, these studies typically involve the pooling of DNA samples from multiple plants (or individuals) and sequencing them together, as seen in previous works (Gaj et al. 2012; Yang et al. 2015; Lirakis et al. 2022). When researchers aim to leverage existing sequence data, such as the 3,000 Rice Genomes (3K RG) dataset, it necessitates bioinformatics skills, particularly proficiency in coding. This requirement for coding expertise can pose a significant challenge for breeders and researchers. As a result, this technical hurdle can be a notable barrier to conducting these studies, particularly within the field of plant breeding”.

Reviewer Comment 10: graphical interface (GUI). Here the 'user' word is missed on page 5 second para

Authors' Response: Thank you for pointing out this error. This sentence has been edited to “graphical user interface (GUI)”

Reviewer Comment 11: Page no 26. The panel of 399 lines is developed based on maturity of 3K lines. How we can say that the Pools for yield per plant are developed based on 3K rice accessions? You must have missed extreme trait representatives from 3K panel, as you don't have accessions for

extremely high and low per plant yield from 3K. It would have been convincing if the 399 panel would have been developed based on per plant yield from 3K rice genomes.

Authors' Response: The panel of rice genotypes was specifically assembled to investigate drought tolerance in the context of India. As opposed to comparing yield across the entire 3,000 Rice Genomes (3K) panel, our focus was on the molecular mechanisms underpinning drought resistance. In other words, we aimed to understand how certain genotypes manage to escape the adverse effects of drought stress by maturing early or at a medium rate. These genotypes are particularly valuable for the development of drought-resilient crops. Therefore, our study concentrated on this subset of lines, and the construction of the 399-line panel was based on a sampling method designed to represent the diversity found within the entire 3K rice genotypes.

Reviewer Comment 12: The R script for determining $K=3$ should be provided in Supplementary for readers.

Authors' Response: The Authors are thankful for this valuable comment. The supplementary codes used for initial filtering and other pre-GWAS analysis are added in the GitHub additionally a section on "Code availability" has been added in the MS on page no. 29, 2nd paragraph, and that will provide all the links to the codes.

"CODE AVAILABILITY"

The source code of the Et-GWAS shinyapp is freely available at <https://github.com/IRRI-South-Asia-Hub/Et-GWAS>. The documentation (Readme file) file is also there to provide the details regarding installation and usage of the R package locally and as web tool. The bash scripts and R functions needed to carry out important pre-association analysis such as quality control of genotypic data and visualization of population stratification (Admixture cluster and principle component analysis) are also present in the same GitHub under the scripts folder".

Reviewer #2

Reviewer Comment 1. My feeling is that the paper is largely solid scientifically but not outstandingly novel. I fear that the invention of ever fancier GWAS-based acronyms is in itself not helpful and that the approach of using stress conditions to access a fuller coverage of the genome has already been postulated and even demonstrated by the Fernie group in Golm.

Authors' Response: We are thankful to the Reviewer for thoroughly reviewing the manuscript and providing valuable suggestions. In our opinion, various GWAS methods have been suggested and

employed in crop science, but their accessibility is often limited to experts in bioinformatics. In many cases, plant breeders are required to possess coding skills to effectively analyze data using these GWAS techniques. Moreover, users frequently work with low-density SNP datasets, like genic SNPs, functional SNPs, or pruned SNPs with strong linkage. In our study we have provided a GUI interphase to bridge the gap between the Breeders who generate the valuable data sets and the computer experts who bring meaning to the generated data sets. Using the GUI interphase there's no need for users to obtain their own computing resources, as we offer a webserver for conducting the analysis. We incorporated SNP dataset options to choose from in the R package.

Reviewer Comment 2. As such I feel that the most novel aspect is the "Unraveling of rare variants in 3000 rice genomes" however to my opinion whilst the title is highly enticing, I am not fully convinced that they actually achieved this.

Authors' Response: The authors sought to address the shortcomings of GWAS, including issues like missing heritability and rare alleles. Simultaneously, they aimed to harness the potential of GWAS for the effective identification of genomic regions linked to the specific trait of interest. The findings of this study are expected to provide valuable assistance to researchers in confidently pinpointing robust QTLs. With the overlapping regions identified between Et-GWAS and conventional GWAS during our analysis, have proven that the QTLs identified through our approach are robust and consistent.

December 5, 2023

RE: Life Science Alliance Manuscript #LSA-2023-02352-TR

Dr. Pallavi Sinha
International Rice Research Institute
South Asia Hub
ICRISAT Campus, Patancheru
Hyderabad, Telangana 502324
India

Dear Dr. Sinha,

Thank you for submitting your revised manuscript entitled "Extreme trait GWAS (Et-GWAS): Unraveling rare variants in the 3000 rice genome.". We would be happy to publish your paper in Life Science Alliance pending final revisions necessary to meet our formatting guidelines.

- please consult our manuscript preparation guidelines <https://www.life-science-alliance.org/manuscript-prep> and make sure your manuscript sections are in the correct order. Please incorporate the Conclusion section into your Discussion section. We do not have a separate Conclusion section.
- please add ORCID ID for corresponding (and secondary corresponding) author--you should have received instructions on how to do so
- please add a callout for each Figure to your main manuscript text (Fig 2a,b,c,d; Fig 4a,b; Fig 5a,b,c,d; FigS1a,b; FigS2a,b; FigS3a,b,c; Fig S7a,b; FigS8a,b; FigS10 a,b,c,d)

A. FINAL FILES:

B. MANUSCRIPT ORGANIZATION AND FORMATTING:

Sincerely,

Reviewer #1 (Comments to the Authors (Required)):

Thanks to authors for addressing the comments. I recommend the acceptance of manuscript.

December 13, 2023

RE: Life Science Alliance Manuscript #LSA-2023-02352-TRR

Dr. Pallavi Sinha
International Rice Research Institute
South Asia Hub
ICRISAT Campus, Patancheru
Hyderabad, Telangana 502324
India

Dear Dr. Sinha,

Thank you for submitting your Research Article entitled "Extreme trait GWAS (Et-GWAS): Unraveling rare variants in the 3000 rice genome.". It is a pleasure to let you know that your manuscript is now accepted for publication in Life Science Alliance. Congratulations on this interesting work.

DISTRIBUTION OF MATERIALS:

Again, congratulations on a very nice paper. I hope you found the review process to be constructive and are pleased with how the manuscript was handled editorially. We look forward to future exciting submissions from your lab.

Sincerely,
